# Linking genotypic and phenotypic changes in the *E. coli* long-term evolution experiment using metabolomics

**John S Favate**[1,2], **Kyle S Skalenko**[1,3], **Eric Chiles**[4], **Xiaoyang Su**[4], **Srujana Samhita Yadavalli**[1,3], **Premal Shah**[1,2]*

[1]Department of Genetics, Rutgers University, Piscataway, United States; [2]Human Genetics Institute of New Jersey, Piscataway, United States; [3]Waksman Institute, Rutgers University, Piscataway, United States; [4]Cancer Institute of New Jersey, New Brunswick, United States

**Abstract** Changes in an organism's environment, genome, or gene expression patterns can lead to changes in its metabolism. The metabolic phenotype can be under selection and contributes to adaptation. However, the networked and convoluted nature of an organism's metabolism makes relating mutations, metabolic changes, and effects on fitness challenging. To overcome this challenge, we use the long-term evolution experiment (LTEE) with *E. coli* as a model to understand how mutations can eventually affect metabolism and perhaps fitness. We used mass spectrometry to broadly survey the metabolomes of the ancestral strains and all 12 evolved lines. We combined this metabolic data with mutation and expression data to suggest how mutations that alter specific reaction pathways, such as the biosynthesis of nicotinamide adenine dinucleotide, might increase fitness in the system. Our work provides a better understanding of how mutations might affect fitness through the metabolic changes in the LTEE and thus provides a major step in developing a complete genotype–phenotype map for this experimental system.

*For correspondence:
premal.shah@rutgers.edu

## eLife assessment

This study presents **convincing** evidence that metabolite levels in *Escherichia coli* bacteria from a long-term evolution experiment have changed in consistent ways, which in turn can be explained by recurrent mutations in regulatory genes that affect enzyme expression levels. The use of high-resolution mass spectrometry measuring bulk metabolite levels, in combination with existing gene expression and DNA sequencing datasets provides **valuable** information linking changes in an organism's genome, transcriptome, and metabolome.

## Introduction

Adaptation is the process by which organisms become fitter for the environment in which they live. While rooted in genetic changes, adaptive evolution can be studied at many levels of molecular and organismal phenotypes (*Dobzhansky, 1964*). An organism's metabolism is one of the fundamental molecular phenotypes supporting its life. It is no surprise, then, that adaptive evolution can proceed through changes in metabolism. Studies have found evidence of adaptation by comparing the metabolomes of closely related organisms. An analysis of metabolic differences between species of *Drosophila* found that these differences can affect lifespan and are sex specific (*Harrison et al., 2022*). In prokaryotes, studies suggest that the genes involved in secondary metabolite production evolve quickly and produce large varieties of phenotypes. For example, strains of *Pseudomonas*

*aeruginosa* that can produces surfactants can better mitigate oxidative stress compared to those that cannot (*Santamaria et al., 2022*). *E. coli* have been shown to evolve repeatable metabolic changes when grown under selection to increase their biomass production and metabolism is often targeted for directed evolution when engineering organisms for industrial uses (*Ibarra et al., 2002*). *E. coli* have also been shown produce coexisting subclones that occupy particular metabolic niches in a lab setting (*Spencer et al., 2008*). Despite these natural and laboratory-based examples, relating genetic, metabolic, and fitness changes is challenging because complete sets of molecular data are often unavailable.

Laboratory evolution experiments provide ample opportunities to study metabolic evolution. The long-term evolution experiment (LTEE) is an experimental evolution system where 12 replicate populations of *E. coli* (designated as Ara+1:6 and Ara-1:6, hereafter referred to as A+1:6 and A-1:6) are propagated in a carbon-limited medium with a 24-hr serial transfer regime (*Lenski et al., 1991*). Begun in 1988, the LTEE populations have evolved for more than 75,000 generations, making it the longest running experiment of its kind. Over time, all 12 populations have continued to adapt to this environment. For example, relative fitness (measured as growth rate relative to the ancestral strain) has increased in parallel across the 12 populations (*Lenski et al., 2015*). Cellular size in each population has also increased (*Grant et al., 2021*; *Philippe et al., 2009*). Parallel genomic changes in the LTEE are well characterized, and while some genes are commonly mutated across the replicate populations, most mutations are unique to each line (*Tenaillon et al., 2016*; *Tenaillon et al., 2012*; *Good et al., 2017*; *Limdi et al., 2022*). Despite the variability at the genetic level, the evolved populations display similar gene expression profiles (*Cooper et al., 2003*; *Favate et al., 2022*). In principle, mutations and changes to gene expression could affect the proteome by altering the function of a protein, the amount of protein produced, or the conditions under which it is translated. Because proteins are the biological catalysts for cellular chemistry, mutations, and gene expression changes may affect the metabolome. In the LTEE, several examples of metabolic changes have been documented. The A-3 population has uniquely evolved the ability to metabolize citrate under aerobic conditions (*Blount et al., 2012*; *Quandt et al., 2015*) and A-2 has produced unique, coexisting ecotypes (*Lenski, 2017*; *Plucain et al., 2014*; *Rozen and Lenski, 2000*). These ecotypes exhibit growth-phase-dependent frequencies and differ in their ability to metabolize acetate, with one ecotype (the L ecotype) specializing on glucose and the other (the S ecotype) on acetate. The transposable element induced loss of the rbs operon occurred early and prevents the evolved lines from growing solely on ribose (*Cooper et al., 2001*). Reduced ability to grow on maltose has occurred through other mechanisms (*Meyer et al., 2010*; *Pelosi et al., 2006*; *Travisano et al., 1995*). Despite glucose being the only source of carbon provided, significant changes in the ability to grow on other carbon sources have occurred (*Leiby and Marx, 2014*). In particular, the mutator lines exhibit reduced performance on carbon sources other than glucose. The evolved lines have increased rates of glucose uptake as well as altered flux through some central carbon metabolism pathways (*Harcombe et al., 2013*).

While specific examples of how genetic changes result in metabolic changes in the LTEE (citrate, acetate, maltose, ribose) exist, a more general survey of the metabolome is lacking. Such a survey would allow a better understanding of the effects of genetic and expression changes on metabolism and provide the ability to test hypotheses from previous data. By integrating genomic (*Tenaillon et al., 2016*), expression (*Favate et al., 2022*), and metabolic datasets, we can relate mutations to expression changes and their effects at the metabolic level. We provide ample evidence for hypotheses that specific genetic changes may exert their fitness-altering effects by changing gene expression and, ultimately, metabolism.

## Results
### Survey of metabolic changes in the LTEE
We used liquid chromatography coupled mass spectrometry (LC/MS) to scan a broad mass range using both positive and negative ionization modes to survey the metabolomes of both ancestral strains and clones from each of the 12 evolved lines of the LTEE at 50,000 generations. We sampled two time points, 2 and 24 hr, to represent exponential and stationary phase samples, each with two biological replicates. It is important to note that the 2 hr time point captures the evolved lines, including A-3, during their growth on glucose. Additionally, the evolved lines reach stationary phase after about

4 hr of growth, with A-3 experiencing growth that levels off slowly over around 14 hr (*Blount et al., 2008*). At our 24 hr time point, the evolved lines (except A-3) have been in stationary phase for around 20 hr. The differences between the early, mid, and late stationary phases are unknown. Metabolite abundances were estimated using normalized peak areas relative to standards as described in the Data processing and description. Many metabolites are detected in both ionization modes, making it unclear which mode most accurately represents the abundance of the compound in a sample. As a result, we used the combination of a compound and the ionization mode it was detected in as a feature of the data.

We begin by surveying the data in a metabolome-wide manner. Because *E. coli* exhibits large physiological differences across growth phases (*Jaishankar and Srivastava, 2017*; *Pletnev et al., 2015*; *Navarro Llorens et al., 2010*), we expected comparisons within a growth phase to be more similar than comparisons across growth phases. Indeed, the distribution of within growth-phase correlations of metabolite abundances is higher than the distribution of across growth-phase correlations (*Figure 1—figure supplement 1A*). Because growth-specific differences have the potential to drown out signals of adaptation, we performed all subsequent analyses in a growth-phase-specific manner.

To identify large-scale differences in metabolite abundances, we performed a principal component analysis (PCA) of metabolite abundances in a phase-specific manner. In the exponential phase, the PCA strongly separates A-3 from the other samples (*Figure 1A*). This is possibly due to the unique ability of A-3 to metabolize citrate. Citrate is present in the medium because it was originally added as a chelating agent. However, mutations in A-3 have allowed it to metabolize citrate under aerobic conditions, giving it access to additional carbon that the other lines do not have (*Turner et al., 2017*; *Blount et al., 2012*). Half of the evolved lines have a mutator phenotype and have at least an order of magnitude higher number of fixed point mutations than the non-mutators (*Tenaillon et al., 2016*). Despite the large differences in the number of mutations that have accumulated, we do not observe a strong effect of mutator status on the separation between evolved lines in the PCA. This is consistent with previous observations of a minimal effect of mutator status on changes in gene expression patterns (*Favate et al., 2022*). The first two principal components are dominated by contributions of nucleoside monophosphates, amino acids, and compounds involved in carbon metabolism, such as glucose and succinate (*Figure 1—figure supplement 3*).

Similar to the exponential phase, A-3 separates from the rest of the evolved lines in a PCA of metabolites in the stationary phase (*Figure 1B*). The primary driver of variation in the first principal component (PC1) is the relative abundances of compounds in A-3 (*Figure 1—figure supplement 4A*). In addition to A-3, A-2 also shows a higher degree of separation from the other lines. Though this experiment was performed with single clones, the 12 flasks in the LTEE are not isoclonal but instead consist of competing subclones that sometimes coexist for extended periods (*Good et al., 2017*). A-2 is the best-understood example, consisting of two major subclones, L and S (*Lenski, 2017*; *Plucain et al., 2014*; *Rozen and Lenski, 2000*). The observed separation might be because the A-2 clone in this experiment comes from the L ecotype, which does not grow as well on acetate compared to the other lines. In contrast to exponential phase PCA, mutator status appears to be a driving factor of PC2 in the stationary phase (*Figure 1—figure supplement 4B*) with mutator lines having lower abundances of nucleoside monophosphates relative to non-mutators during stationary phase. What this is indicative of is unclear.

A common theme in studies of LTEE is the high degree of parallelism observed at the genetic (*Tenaillon et al., 2016*), gene expression (*Favate et al., 2022*), and fitness levels (*Wiser et al., 2013*). For example, genes involved in flagellar biosynthesis and amino acid metabolism are commonly mutated across the evolved lines (*Tenaillon et al., 2016*) with associated changes in expression levels (*Favate et al., 2022*). To examine the extent of parallel changes in metabolites, we calculated the ratio of the peak area in an evolved line to the peak area in the ancestor for each compound after averaging biological replicates. Pairwise Spearman's correlations using these ratios are significantly higher than expected when comparing the observed distribution to a distribution generated by randomizing fold-changes within a sample prior to calculating correlations (*Figure 1C*, $p < 0.001$, *t*-test). In the exponential phase, the lowest correlations were mainly between a pair involving one of A-2, A-4, or A-6, and another evolved line (*Figure 1D*). The correlations of metabolite abundances across evolved lines are more similar in the exponential phase (*Figure 1D*) than in the stationary phase (*Figure 1E*) (*t*-test comparing the observed exponential and stationary phase correlations in *Figure 1C*, $p < 0.001$).

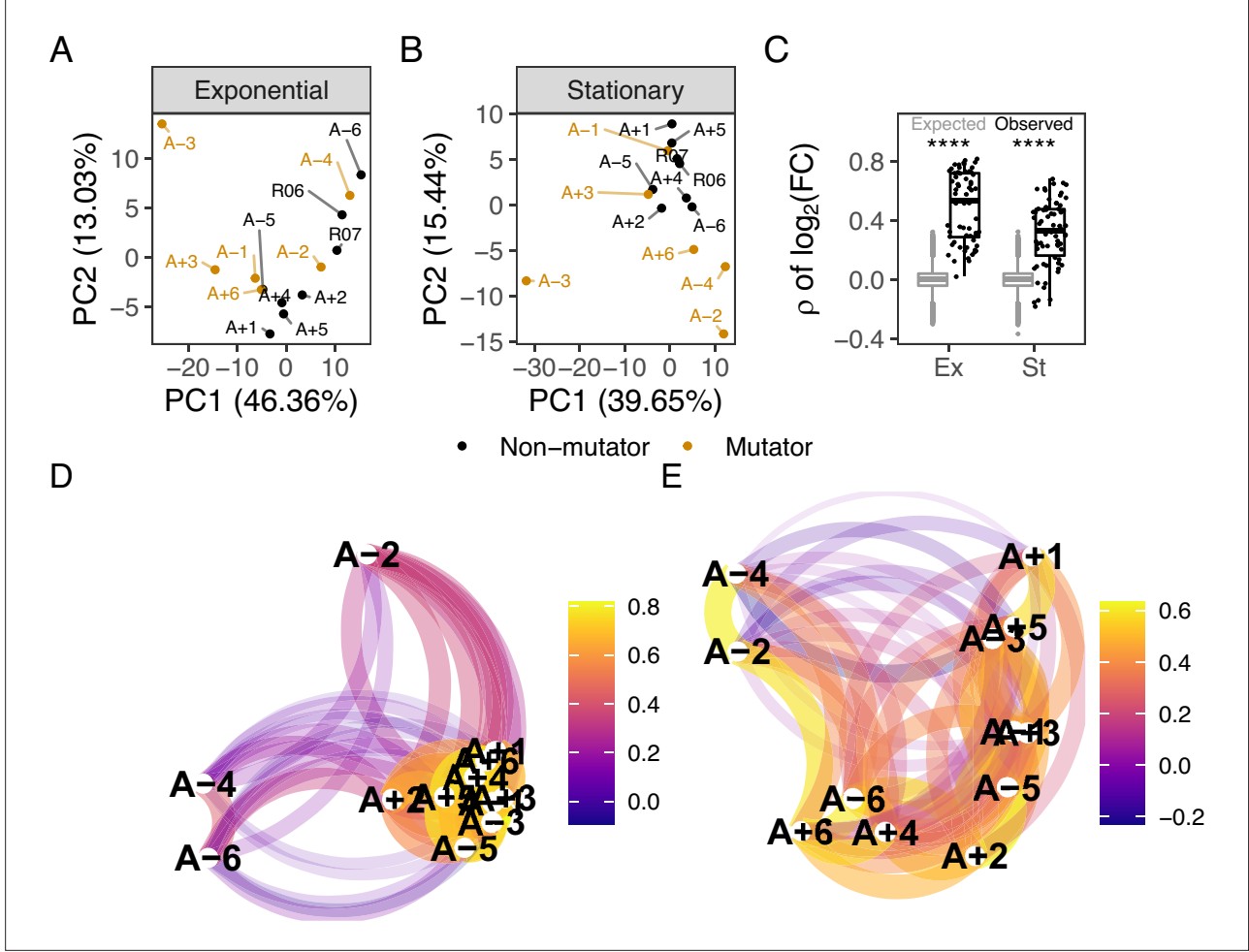

**Figure 1.** Comparison of metabolic changes in evolved lines within each growth phase. (**A, B**) Principal component analysis based on $log_{10}(mean\ normalized\ peak\ area)$ separated by growth phase. R06 and R07 are the ancestors (REL606 and REL607). For this figure, the combination of ionization mode and metabolite was treated as a feature of the data. (**C**) Pairwise Spearman's correlations based on $log_2(fold-change)$ relative to the ancestor. The black boxes and points indicate the observed correlations, the gray boxes indicate correlations calculated after 100,000 randomizations of fold-changes within each line. p-values indicate the results of a two-tailed $t$-test between the observed and expected distributions. **\*\*\*\*** indicates a p-value ≤0.0001. (**D, E**) The observed correlations from C plotted in a network manner. (**D**) is the exponential phase and (**E**) is the stationary phase. Lines are clustered based on similarity and the color of the line connecting two points indicates the strength of the correlation.

The online version of this article includes the following figure supplement(s) for figure 1:

**Figure supplement 1.** Pairwise comparisons of mass-spectrometry data across growth phases.

**Figure supplement 2.** Distributions of peak areas for compounds whose values were imputed using a quantile regression imputation of left-censored (QRILC) method (see Data processing and description for a complete description).

**Figure supplement 3.** Relationship between key metabolites impacting principal components of exponential phase metabolomes of evolved lines.

**Figure supplement 4.** Relationship between key metabolites impacting principal components of stationary phase metabolomes of evolved lines.

**Figure supplement 5.** The theoretical and observed probabilities of finding features (the combination of metabolite and the ionization mode it was detected in) that are significantly altered ($|log_2(fold-change)| \geq 1$) in a given number of evolved lines (x-axis).

**Figure supplement 6.** The theoretical and observed number of shared, significantly altered ($|log_2(fold-change)| \geq 1$) metabolic features (the combination of metabolite and the ionization mode it was detected in) in a given number of evolved lines (x-axis).

It may be that the strategies of metabolite usage and abundance in the rapidly growing exponential phase are more similar across lines than the strategies of survival are in the stationary phase or that there is less selection on the metabolome during the stationary phase. Further studies would be required to understand this phenomenon.

To further quantify the extent of shared changes, we considered the difference between the number of shared changes that were observed and expected to be observed by chance. We approximated an expected distribution of shared changes using a Sum of Independent Non-Identical Binomial Random Variables (SINIB) method (*Liu and Quertermous, 2018*), essentially asking what the chance of repeatedly observing the same change in an increasing number of evolved lines is. For more details on this method, see Theoretical distributions for parallel changes in metabolites. For metabolic features (the combination of a compound and the ionization mode it was detected in) that shared alterations in only a few lines (generally six or less), we cannot suggest selection on these changes because we see fewer or as many changes as expected (*Figure 1—figure supplement 5*, *Figure 1—figure supplement 6*). These may represent metabolites that negatively affect fitness when altered. On the other hand, we observe more changes shared in a higher number of evolved lines than expected. These shared changes may be beneficial, but this would need to be clarified by additional experiments. By combining genomic, expression, and metabolic datasets, we can better explore the nature of these changes and how they may impact fitness.

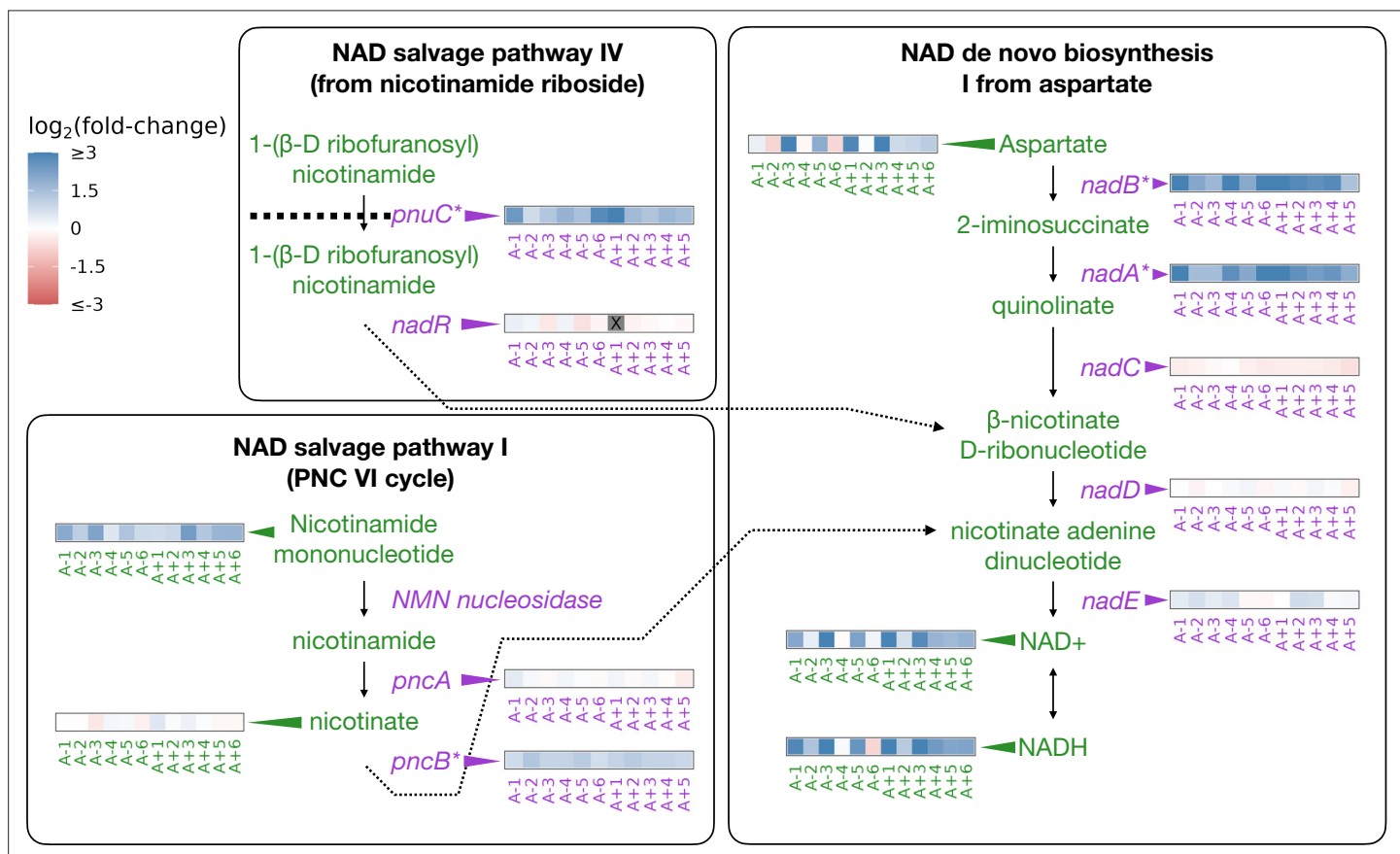

**Figure 2.** Depiction of three pathways (bold-faced text) that contribute to NAD abundances in the cell. Graphics and pathway names are adapted from the EcoCyc database (*Keseler et al., 2005*). All data represent exponential phase measurements. Genes that code for enzymes are shown in purple and metabolites in green. Heatmaps positioned to the right of gene names show the fold-change in expression relative to the ancestor (data from *Favate et al., 2022*). Gray spaces (also marked with an X) in gene expression heatmaps represent evolved lines where that gene contains an indel or is deleted. Asterisks indicate genes that are transcriptionally regulated by NadR. Heatmaps positioned to the left of metabolite names show changes in metabolite abundance relative to the ancestor. PnuC transports compounds into the cell. Each heatmap represents one ionization mode, but a mixture of positive and negative ionization mode data is shown depending on which mode a compound was detected in. See *Figure 2—figure supplement 1A* for complete data.

The online version of this article includes the following figure supplement(s) for figure 2:

**Figure supplement 1.** Relationship between metabolites of the NAD pathway across evolved lines.

## Increased nicotinamide adenine dinucleotide may facilitate higher energy demands during adaptation

In the LTEE, mutations to the nadR gene have occurred in all of the evolved lines and appear to have been under strong selection to improve fitness (*Woods et al., 2006*; *Tenaillon et al., 2016*). NadR is a dual-function protein that negatively regulates genes involved in the synthesis of nicotinamide adenine dinucleotide (NAD) and has its own kinase activity (*Begley et al., 2001*; *Osterman, 2009*).

NAD and its phosphorylated derivative NADP function as electron carriers and are necessary for many catabolic and anabolic pathways in the cell. While untested, hypotheses for how nadR mutations might increase fitness involve derepression of its target genes and increases to intracellular NAD; this may aid in shortening the lag phase and modulating redox chemistry (*Woods et al., 2006*; *Grose et al., 2005*; *Grose et al., 2006*). The abundance of NAD and related compounds in a cell can modulate reaction rates and other processes (*Cantó et al., 2015*; *Osterman, 2009*). We had previously shown that mutations in nadR in the evolved lines lead to consistent upregulation of its target genes (*Favate et al., 2022*; *Figure 2*). Here, we test if there are higher abundances of NAD and NAD-related metabolites in the evolved lines consistent with changes observed at the genetic and gene expression levels.

The genes regulated by NadR participate in specific reactions along NAD synthesis pathways. We detected some of the compounds in these pathways as well as both oxidized and reduced forms of NAD and NADP (*Figure 2*). Compared to the ancestors, both redox states of NAD and NADP are almost universally increased in the evolved lines during the exponential phase (*Figure 2*, *Figure 2—figure supplement 1A*), with median fold-changes of 3.32 and 4.65 for NAD and NADH, respectively. NADP is generated through phosphorylation of NAD by NadK (*Kawai et al., 2001*; *Osterman, 2009*) and saw median fold-changes of 2.34 and 3.54 for NADP and NADPH, respectively. Because NADP is generated from NAD, and each exists in two oxidation states, we expected all four compounds to see similar increases within an evolved line. Indeed, increases in the various redox and phosphorylation states were consistent within an evolved line ($0.77 < R < 0.97$, *Figure 2—figure supplement 1B*). Aspartate, the starting point for NAD synthesis, was also increased in most of the evolved lines, with a median fold-change of 1.94. Nicotinamide mononucleotide, which can be used to make NAD, was consistently increased, with a median fold-change of 2.51. In the stationary phase, most of these patterns remain unchanged except for aspartate, which is generally lower in abundance compared to the ancestor (*Figure 2—figure supplement 1A*). These results suggest that the end product of the mutations in nadR is increased intracellular NAD.

## Derepression of arginine biosynthesis at genetic, gene expression, and metabolic levels

Similar to nadR, another consistent target of mutations in the LTEE is a transcriptional repressor of genes involved in arginine biosynthesis argR (*Tenaillon et al., 2016*). ArgR, along with arginine, negatively autoregulates the arginine biosynthesis pathway by downregulating the participating genes when arginine is abundant (*Tian and Maas, 1994*). As an amino acid, the primary role of arginine is in protein synthesis. However, various reactions produce and consume arginine; both its synthesis and the regulation of the enzymes involved are complex (see *Charlier and Glansdorff, 2004* and the EcoCyc *Keseler et al., 2005* pathway 'superpathway of arginine and polyamine biosynthesis'). For example, arginine can be a source of nitrogen and carbon or be used for ATP synthesis (*Reitzer, 2005*). Because of these complications, if mutations in argR and the subsequent changes in gene expression do affect this pathway, it is unclear at which step one might observe a change or if that change might be in a different pathway.

Mutations in argR are localized to specific regions, potentially disrupting the DNA-binding domain, the arginine sensing domain, or interaction domain, which is required for hexermerization of the protein (*Tian and Maas, 1994*). These mutations in argR could reduce its ability to repress its target genes, resulting in their increased expression. Consistent with these expectations, we had previously observed parallel increases in the expression of genes repressed by ArgR (*Figure 3*; *Favate et al., 2022*).

We were able to detect 11 compounds involved in the superpathway of arginine and polyamine biosynthesis (*Figure 3*, *Figure 3—figure supplement 1A*). Compared to the ancestor, arginine was increased in nine (negative mode) or ten (positive mode) evolved lines and experienced a median

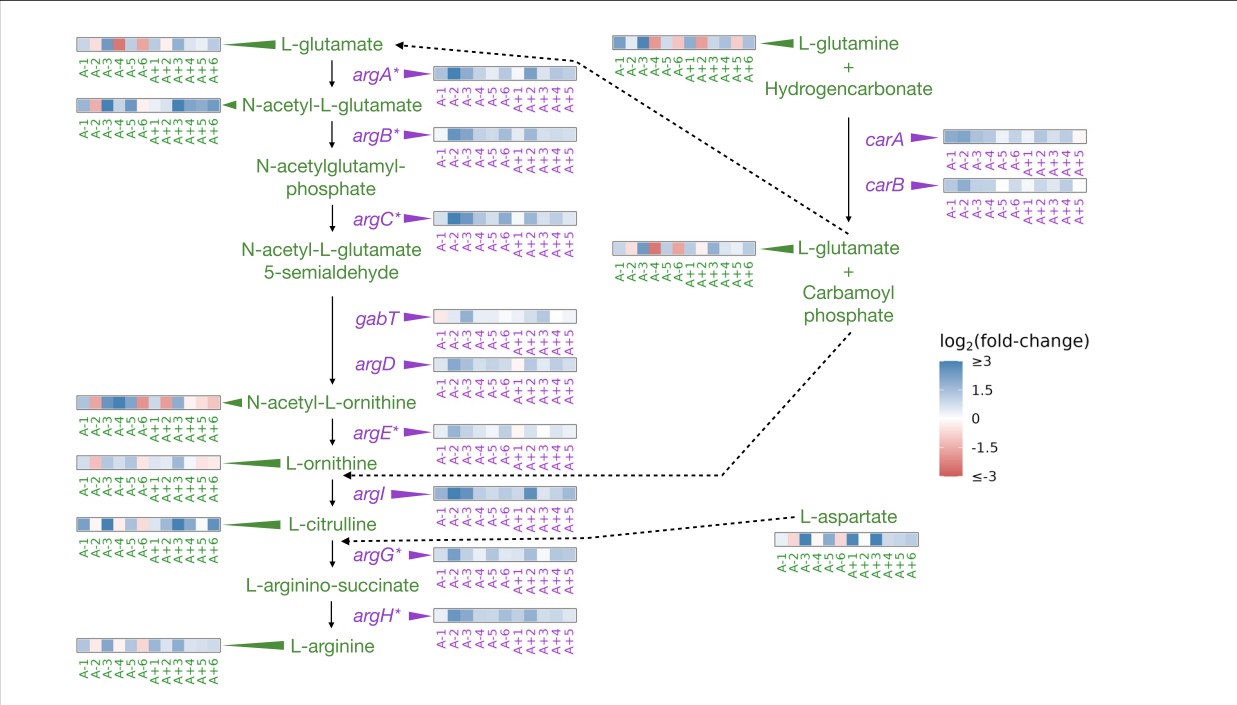

**Figure 3.** Partial depiction of the pathway 'superpathway of arginine and polyamine biosynthesis'(**Keseler et al., 2005**). All data represent exponential phase measurements. Genes that code for enzymes are shown in purple, and metabolites in green. Heatmaps positioned to the right of gene names show the fold-change in expression relative to the ancestor (data from **Favate et al., 2022**). Asterisks indicate genes that are transcriptionally regulated by ArgR. Heatmaps positioned near metabolite names show changes in metabolite abundance relative to the ancestor. Each heatmap represents one ionization mode, but a mixture of positive and negative ionization mode data is shown depending on which mode a compound was detected. See *Figure 3—figure supplement 1A* for complete data and *Figure 3—figure supplement 2* for line-specific data.

The online version of this article includes the following figure supplement(s) for figure 3:

**Figure supplement 1.** Changes in metabolite abundances of the Arginine biosynthesis pathway across evolved lines.

**Figure supplement 2.** Evolved line-specific metabolite and expression changes in Arginine biosynthesis pathways.

fold-change of 1.9 across both ionization modes in the exponential phase. This pattern also persisted in the stationary phase (*Figure 3—figure supplement 1A*). While arginine is increased in both growth phases in most of the evolved lines, other compounds in this pathway show highly variable changes (*Figure 3—figure supplement 1A*). Despite the consistent relationship between mutations, expression changes, and changes in the abundance of arginine, how this might affect fitness is not obvious. Higher intracellular abundances of amino acids could facilitate higher translation rates and promote faster growth. While we do observe that some amino acids show increases in their abundances in most evolved lines in the exponential phase (*Figure 3—figure supplement 1B*), further experiments would be needed to confirm if these amino acids are used in protein synthesis or in other pathways.

## Functional changes in the central carbon metabolism

Low carbon availability is the key feature of the minimal medium used in the LTEE, and selection for better use of carbon is a driving force of adaptation in the system. Hence, how mutations and expression changes might affect fitness by altering central carbon metabolism pathways is of interest. The evolved lines have seen significant positive and negative changes in their ability to grow on different carbon substrates. Rather than metabolic specialization, this was found to be due to the accumulation of deleterious mutations that affect the ability to grow on other substrates (**Leiby and Marx, 2014**). Unique changes in A-3 allow it to metabolize citrate under aerobic conditions, granting it access to extra carbon from an unused ecological niche (**Blount et al., 2012**; **Quandt et al., 2015**). The evolved lines retain more carbon in their biomass compared to the ancestors (**Turner et al., 2017**), and increased glucose uptake rates have been demonstrated (**Harcombe et al., 2013**). Despite the abundance of molecular data associated with the LTEE, the highly networked nature of the genes

and metabolites involved in central carbon metabolism makes relating the various molecular data to each other challenging. In particular, mass spectrometry data alone will not allow us to differentiate between sources for compounds that are generated by many reactions. We might overcome this limitation by combining genomic, expression, and metabolic datasets.

We again sought instances of parallel sets of mutations and expression changes that may exert their effects at the metabolic level. Three genes that are key components in the glyoxylate cycle, aceB, aceA, and aceK were commonly upregulated in the evolved lines (*Favate et al., 2022*). Their transcriptional regulators iclR and arcB are heavily mutated (*Tenaillon et al., 2016*), and this is known to be beneficial in the LTEE (*Quandt et al., 2015*). The glyoxylate cycle allows the use of acetate as a carbon source by converting it to succinate, which can later be fed into the citric acid cycle (*Keseler et al., 2005*; *Cioni et al., 1981*). Because these genes had increased expression, the compounds they produce (succinate, glyoxylate, and malate) may be elevated in the evolved lines relative to the ancestor. Succinate and malate were reliably identified in our data, but glyoxylate was not. Unfortunately, acetate was part of the mobile phase of our liquid chromatography setup, thus preventing its accurate quantification. Likewise, most of the molecules in the glyoxylate cycle, including succinate and malate, are also components of the citric acid cycle and can be produced or consumed by other enzymes. Because we cannot distinguish where changes in these metabolites come from, we chose to look at compounds from central carbon metabolism in general, considering compounds from the glyoxylate cycle, glycolysis, and the citric acid cycle.

Overall, 18 compounds from these pathways are detected in our LC/MS assay. Regarding the glyoxylate shunt, malate, aconitate, and succinate were generally elevated in the evolved lines (median fold-change of 5.38, 1.66, and 2.30, respectively (*Figure 4*)). Interestingly, the evolved lines appear to have similar or lower amounts of glucose despite having been shown to have increased glucose uptake rates (*Harcombe et al., 2013*). This discrepancy is likely due to the fact that in *Harcombe et al., 2013*, glucose uptake rates were calculated by measuring the depletion of glucose in the medium, whereas we measured glucose from inside the cells. Once inside the cell, we suspect that the glucose is quickly used. This is supported by the fact that other downstream glycolysis compounds, like phosphoenolpyruvate (PEP), are generally elevated (median fold-change of 6.72, *Figure 4*).

An increase in PEP is also consistent with a hypothesis related to increased glucose uptake. It was previously noted that all of the evolved lines contain what are likely inactivating mutations in the gene pykF. pykF encodes pyruvate kinase I, one of two isozymes that generate ATP by converting PEP to pyruvate (*Philippe et al., 2007*). It was thought that inactivating mutations in pykF reduce the conversion of PEP to pyruvate, forcing a buildup of PEP, and that because PEP is the energy source for glucose import in the sugar-transporting phosphotransferase system, a buildup of PEP might increase glucose uptake rates. Our data, which shows increases in PEP, lend support to these hypotheses.

The citric acid cycle is a major metabolic pathway that extracts energy via the reduction of electron carriers like NAD. These electrons are shuttled to the electron transport chain by NAD, where they are used to generate ATP. NAD can be reduced at two points during the citric acid cycle, the conversion of alpha-ketoglutarate to succinate and the conversion of malate to oxaloacetate. If the citric acid cycle is running faster in the evolved lines due to their higher growth rates and hence higher energy demands, then more NAD may be required to efficiently shuttle electrons from the citric acid cycle to the electron transport chain. Interestingly, we measured a median fold-change of 3.13 for alpha-ketoglutarate and 5.38 for malate (*Figure 4*). Additionally, increases in these compounds are correlated with an increase in NAD compounds within an evolved line (*Figure 4—figure supplement 1*). It may be that increases in NAD allow faster operation of the citric acid cycle. However, alternative hypotheses must be considered.

Overflow metabolism appears to be common in the evolved lines. Overflow metabolism refers to the seemingly wasteful production of metabolites that do not participate in the most efficient form of ATP generation, even when glycolytic substrates and oxygen are plentiful. The Warburg effect (*Warburg, 1956*) and acetate production are examples of this phenomenon in human cancers and *E. coli* (*Enjalbert et al., 2017*), respectively. Our observation of the buildup of PEP is an example of overflow metabolism and may lend support to the hypothesis that the evolved lines are using overflow metabolism to gain access to the glycolytic intermediates and using them for other purposes, like as an energy source for sugar import. Experiments using radioactive tracers to study flux through these pathways are needed to confirm any hypotheses.

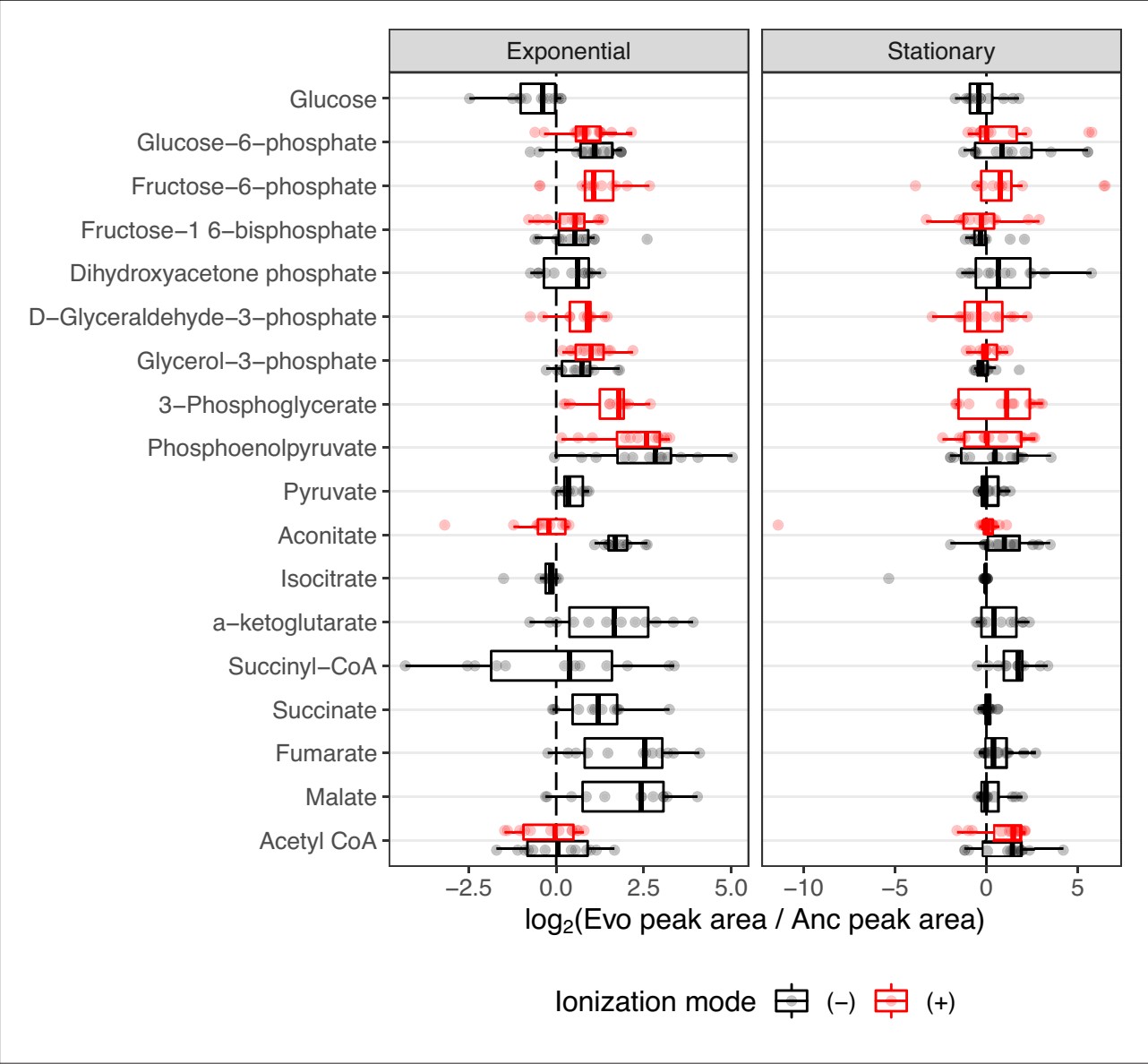

**Figure 4.** The distribution of fold-changes relative to the ancestor for compounds involved in carbon metabolism. Red and black indicate detection in positive or negative ionization mode, respectively. Not all compounds were detected in both ionization modes. Compounds are ordered from top to bottom roughly as they occur in glycolysis or other reactions.

The online version of this article includes the following figure supplement(s) for figure 4:

**Figure supplement 1.** The relationship between fold-changes in malate and α-ketoglutarate and NAD(H) are correlated within an individual evolved line.

## Discussion

The metabolome of a cell is the integration point of an organism's environment, genetics, and gene expression patterns. Metabolism has been shown to contribute to generating adaptive phenotypes (**Harrison et al., 2022**; **Miyazawa and Noguchi, 2001**; **Hefetz and Blum, 1978**; **Chevrette et al., 2020**). However, how mutations affect metabolism through changes in gene expression and ultimately affect fitness is less clear. Specific examples of this have been studied in the LTEE, such as the acquisition of citrate metabolism in A-3 (**Blount et al., 2012**; **Quandt et al., 2015**). We used metabolic mass spectrometry to study other aspects of the LTEE, relating mutations and expression changes to metabolic changes. This allowed us to lend support to previous hypotheses of how certain mutations affect fitness in the system. For instance, we show how mutations in nadR alter the expression of its

target genes and increase the intracellular abundance of NAD. It also enables new areas of investigation, such as what role mutations to argR and subsequent expression and metabolic changes might mean for the system.

A key limitation of the data presented here is that it contains measurements of 196 metabolites out of the 3755 metabolites currently annotated in the *E. coli* metabolome database (*Sajed et al., 2016*). Nonetheless, analysis of this subset of data can reveal some global patterns of metabolite changes. For example, distinct clustering of A-3 and A-2 in the PCAs suggests that even with a limited sampling of the metabolome, the unique characteristics of these lineages are observable at the metabolic level. Interestingly, while the mutator lines do not form a well-defined group in the exponential phase, their reduced abundances of nucleoside monophosphates in the stationary phase group them together (*Figure 1B*). Due to deletions of key biosynthetic genes, the mutators are less flexible than the non-mutators in their ability to grow on different carbon sources (*Leiby and Marx, 2014*). A-6, in particular, has one of the higher mutational burdens and is the least flexible in its ability to grow on different carbon substrates. Pathways with missing or broken genes may cause the shunting of those compounds to different pathways while also starving downstream reactions of the now missing reactants.

Though most studies of the LTEE focus on the activity of the evolved lines in the exponential phase, they spend most of their time in the stationary phase. Experiments studying the evolution of *E. coli* during long-term stationary phase (around 3 years of culture with no addition of resources) showed that mutations continue to accumulate during this time and that patterns suggestive of adaptive evolution had occurred (*Katz et al., 2021*; *Ratib et al., 2021*). While our study chose the latest possible time point for examining the stationary phase (immediately before the transfer to a new flask), differences between early, mid, and late stationary phases may exist. Time course experiments with a finer resolution would reveal details about the specific nature of the stationary phase in the LTEE.

## Methods
### Cell culture

The A-3 clone used in this experiment is capable of metabolizing citrate, and the clone of A-2 is the L variant (*Favate et al., 2022*). Frozen bacterial stocks were revived by growing them in 5 ml LB broth for 24 hr. Following this, 1% of each culture was transferred to 5 ml standard LTEE medium for another 24 hr. After 24 hr, these cultures were used to initiate the final experimental cultures. Each clone was grown in 250 ml of medium in a 1-l flask following standard LTEE protocols. The medium used was standard LTEE medium with the standard amount of glucose, 25 mg/ml. After 2 hr of growth, 160 ml of the sample was removed for exponential phase mass spectrometry samples. After 24 hr of growth, 40 ml of the sample was removed for stationary phase samples. Each line had two independent biological replicates.

### Mass spectrometry sample collection

Cells were collected via vacuum filtration using Millipore Omnipore 0.2 μm filters (JGWP04700). Sterile plastic Petri dishes were placed onto a metal tray on dry ice, and 1.2 ml of extraction solvent (40:40:20 acetonitrile:methanol:water +0.5% formic acid) was added to the Petri dish to chill. When filtration was complete, the filter was placed cell-side down into the Petri dish. The metal tray was moved to wet ice to extract for 20 min. After 20 min, the acid was neutralized by the addition of 1.07% (final) ammonium bicarbonate, then cell debris was spun down at 14,000 rpm at 4°C for 10 min, and the clarified extract was transferred to a chilled 2 ml tube and stored at −20°C. The filter was extracted again by the addition of 0.4 ml extraction solvent and sat for 15 min. The extract was neutralized, clarified, and consolidated with the first extract and incubated overnight at −20°C.

The next day, precipitated protein was spun down at 14,000 rpm at 4°C for 10 min, and solvents were removed by speed vac for 2 hr at room temperature. Lastly, the sample was concentrated by removing the remaining water by lyophilization. Dried samples were reconstituted in 40 μL of extraction solvent and submitted for mass spectrometry.

## Mass spectrometry

Mass spectrometry was performed as described previously in *Su et al., 2020* at the Rutgers University Cancer Institute of New Jersey metabolomics core facility using a Thermo Q Exactive PLUS coupled with a Thermo Vanquish UHPLC system.

## Data processing and description

The raw mass spectrometry data are deposited at the metabolomics workbench under the study ID ST002431 and the code for the analysis at https://github.com/shahlab/ltee_massspec, (copy archived at *Favate, 2023*). Normalization of the mass spectrometry data was performed in two steps. First, raw peak areas were normalized against internal standards as in *Su et al., 2020*. After these values were generated, all other data processing steps were performed using the R programming language (*R Development Core Team, 2022*) and the tidyverse set of packages (*Wickham et al., 2019*). Code for this section can be found in the document titled data_processing.Rmd. Peak areas resulting from the first step of normalization were then taken as a proportion of the total peak area for a single sample to normalize against differences in input amounts.

As noted in the text, not all of the 196 compounds were detected in all samples. This varied depending on the compound, line, growth phase, and ionization mode. Overall, 168 compounds were present in all samples in at least one ionization mode, with the remaining 28 compounds being undetected in at least one of the samples in at least one of the ionization modes. We used methods from *Wei et al., 2018* and the accompanying R package imputeLCMD to evaluate different imputation methods for imputing missing data, settling on the quantile regression imputation of left-censored (QRILC) data method. Imputed values theoretically represent values below the limit of detection rather than a complete absence of compounds. As expected, our imputed values always fell below the detected values for a given compound (*Figure 1—figure supplement 2*). After imputation was performed, correlations between the replicates were high (*Figure 1—figure supplement 1A*). Compounds that were detected in both ionization modes show a modest correlation within a sample (*Figure 1—figure supplement 1B*). Distributions of normalized peak areas are similar across replicates and samples (*Figure 1—figure supplement 1C*). This completed dataset was used for further analysis and is available as *Supplementary file 1*. Where appropriate, we show data from both ionization modes, treating the combination of a compound and the ionization mode it was detected in as a feature of the data and compare these across samples.

## Theoretical distributions for parallel changes in metabolites

The code for this analysis can be found in the document titled parallelism.Rmd. We used the R package SINIB (*Liu and Quertermous, 2018*) to calculate theoretical probabilities of finding a shared metabolic change in a particular number of evolved lines. First, we designated metabolic features (the combination of compound and the ionization mode it was detected in) as significant if they experienced an $|log_2(fold-change)| \geq 1$. Then, we determined the probability of randomly picking a significantly altered metabolic feature in each evolved line in a manner specific to the growth phase and direction of change. We then parameterized the dsinib function with these probabilities, essentially asking what the chance of repeatedly finding the same change in an increasing number of evolved lines is. We used these probabilities to determine the total number of metabolic features ones might expect to find altered in the same direction. These numbers can be compared to the observed distribution, showing that more parallel changes are observed than expected.

## Acknowledgements

We thank Richard Lenski for generously providing clones from the LTEE. Premal Shah is supported by NIH/NIGMS grant R35 GM124976 and start-up funds from the Human Genetics Institute of New Jersey at Rutgers University. Srujana S Yadavalli is supported by NIGMS R35 GM147566 and institutional start-up funds. Mass spectrometry data were generated by the Rutgers Cancer Institute of New Jersey Metabolomics Shared Resource, supported in part with funding from NCI-CCSG P30CA072720-5923.

## Additional information

### Competing interests

Kyle S Skalenko: Kyle S. Skalenko is a scientist at Specialty Assays Inc. Srujana Samhita Yadavalli: Srujana S Yadavalli consults and collaborates with Designs for Vision Inc. Premal Shah: Premal Shah is a member of the Scientific Advisory Board of Trestle Biosciences and is a Director at Ananke Therapeutics. The other authors declare that no competing interests exist.

### Funding

| Funder | Grant reference number | Author |
| --- | --- | --- |
| National Institutes of Health | R35 GM124976 | Premal Shah |
| National Institutes of Health | R35 GM147566 | Srujana Samhita Yadavalli |
| National Institutes of Health | CCSG P30CA072720-5923 | Xiaoyang Su |
| Rutgers, The State University of New Jersey | | Srujana Samhita Yadavalli Premal Shah |
| Rutgers Cancer Institute of New Jersey | | Xiaoyang Su |

The funders had no role in study design, data collection, and interpretation, or the decision to submit the work for publication.

### Author contributions

John S Favate, Resources, Data curation, Software, Formal analysis, Validation, Investigation, Visualization, Methodology, Writing – original draft; Kyle S Skalenko, Data curation, Validation, Investigation, Methodology; Eric Chiles, Formal analysis, Investigation, Methodology; Xiaoyang Su, Data curation, Formal analysis, Supervision, Investigation; Srujana Samhita Yadavalli, Data curation, Supervision, Funding acquisition, Investigation, Methodology, Writing – review and editing; Premal Shah, Conceptualization, Resources, Formal analysis, Supervision, Funding acquisition, Investigation, Visualization, Writing – original draft, Project administration, Writing – review and editing

### Author ORCIDs

John S Favate http://orcid.org/0000-0001-6344-4854
Premal Shah https://orcid.org/0000-0002-8424-4218

Reviewer #1 (Public Review): https://doi.org/10.7554/eLife.87039.3.sa1
Reviewer #2 (Public Review): https://doi.org/10.7554/eLife.87039.3.sa2
Author Response https://doi.org/10.7554/eLife.87039.3.sa3

## Additional files

### Supplementary files

• MDAR checklist

• Supplementary file 1. Mass-specrometry data. This table contains the finalized data using for the analysis, including the imputed values. It contains the following columns; charge - the ionization mode of the instrument; line - the name of the strain; phase - (e)xponential or (s)tationary growth phase; repl - replicate; compound - name of the compound; is_standard- was this compound a calibratrion standard?; was_imputed - was this value imputed?; peak_area - the raw peak area; n_peak_area - the normalized peak area.

## Data availability

The raw mass spectrometry data are deposited at the metabolomics workbench under the study ID ST002431 and the code for the analysis at https://github.com/shahlab/ltee_massspec, (copy archived at *Favate, 2023*).

The following dataset was generated:

| Author(s) | Year | Dataset title | Dataset URL | Database and Identifier |
|---|---|---|---|---|
| Favate JS, Shah P | 2022 | MS profiling of the Long Term Evolution Experiment | https://doi.org/10.21228/M8J41M | metabolomicsworkbench, 10.21228/M8J41M |

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
