## [Editor Report · eLife assessment]

This study presents **convincing** evidence that metabolite levels in *Escherichia coli* bacteria from a long-term evolution experiment have changed in consistent ways, which in turn can be explained by recurrent mutations in regulatory genes that affect enzyme expression levels. The use of high-resolution mass spectrometry measuring bulk metabolite levels, in combination with existing gene expression and DNA sequencing datasets provides **valuable** information linking changes in an organism's genome, transcriptome, and metabolome.

---

## [Referee Report · Reviewer #1 (Public Review)]

Summary:

Favate et al. measure the relative levels of metabolites in 12 **Escherichia coli* *strains isolated from different replicate populations after 50,000 generations of the Lenski long-term laboratory evolution experiment. They use untargeted LC/MS methods that include standards and report both positive and negative ionization mode measurements. They initially use principal component analysis (PCA) to broadly compare how the metabolomes of these strains are similar and different. Then, they describe several instances where the changes in metabolite abundance they see in specific pathways correlate with mutations that lead to changes in the expression of genes that encode enzymes in those pathways.

Strengths:

The statistical analyses and presentation of the high-throughput data are excellent. The most compelling results are communicated in wonderful figures that integrate their measurements of metabolite levels in this study with results from a prior study they conducted looking at changes in gene expression levels in the same bacterial strains. These sections include the ones describing large increases in NAD(P) pools due to mutations in nadR, changes in the levels of arginine and related compounds due to mutations in argR, and changes in metabolites from glycolysis and the TCA cycle related to iclR and arcB.

Weaknesses:

After addressing prior reviews, the main remaining weaknesses of the study are limitations inherent to the metabolomics approach that are noted by the authors. Namely, that it gives a static and incomplete picture of cellular metabolism, lacking any information about flux and missing measurements for many metabolites. Additional biochemical and genetic experiments will be necessary to fully test the hypotheses suggested by the metabolomics data.

Impact and Significance:

While there has been past speculation about the effects of LTEE mutations on metabolism, this study measures changes in the levels of metabolites in related metabolic pathways for the first time. Therefore, it provides useful information about how metabolism evolves, in general, and will also be a useful resource those studying other aspects of the LTEE related to metabolism, such as contingency in the evolution of citrate utilization.

---

## [Referee Report · Reviewer #2 (Public Review)]

This preprint presents a compelling study examining the relationship between genotypic changes and phenotypic traits in bacteria over an extended period using the Long-Term Evolution Experiment (LTEE) as a model. The primary advances in methodology include employing high-resolution mass spectrometry for comprehensive metabolic profiling and combining it with previous gene expression and DNA sequencing datasets. This approach provides insight into how specific genetic mutations can alter metabolic pathways over 50,000 generations, enabling a deeper understanding of how genetic changes lead to observed differences in evolved bacterial strains. The findings reveal that evolved bacteria possess more diverse metabolic profiles compared to their ancestors, suggesting that these populations have uniquely adapted to their environment. The work also attempts to uncover the molecular basis for this adaptive evolution, demonstrating how specific genetic changes have influenced the bacteria's metabolic pathways.

Overall, this is a significant and well-executed research study. It offers new insights into the complex relationship between genetic changes and observable traits in evolving populations and utilizes metabolomics in the LTEE, a novel approach in combination with RNA-seq and mutation datasets.

---

## [Author Response]

The following is the authors’ response to the original reviews.

**Reviewer #1 (Public Review):**
[...] WeaknessesShowing that A-2 and especially A-3 are outliers in the PCA analysis is useful, but it may be hiding other interesting signals in the data. The other strains are remarkably colinear on these plots, hinting that if the outliers were removed, one main component would emerge along which they are situated. It also seems possible that this additional analysis step would allow the second dimension to better differentiate them in a way that is interesting with respect to their mutator status or mutations in key metabolic or regulatory genes.

We thank the reviewer for their positive comments and their constructive feedback on the manuscript. Following reviewer’s recommendation, we performed the PCA analysis on metabolism data after removing A-2 and A-3 data. We have detailed those results below. Consistent with a similar analysis performed on RNA-seq datasets in our previous publication, we find that removing these outliers has only a modest effect on separating mutators from non-mutators. We find that, while the new PC2 separates most mutators from the non-mutators, the separation is rather weak. Moreover, we do not see a similar distinction when looking at metabolic data in the Stationary phase. In the interest of improving the readability of the manuscript, we recommend not including these analysis in the final manuscript. We have presented the data for the reviewer’s benefit in Author response image 1, 2 and 3.

**Author response image 2. sa3fig2:** 

**Author response image 3. sa3fig3:** 

There is a missed opportunity to connect some key results to what is known about LTEE mutations that reduce the activity of pykF (pyruvate kinase I). This gene is mutated in all 12 LTEE populations, and often these mutations are frameshifts or transposon insertions that should completely knock out its activity. At first glance, inactivating an enzyme for a step in glycolysis does not make sense when the nutrient source in the growth medium is glucose, even though PykF is only one of two isozymes *E. coli* encodes for this reaction. There has been speculation that inactivating pykF increases the concentration of phosphoenolpyruvate (PEP) in cells and that this can lead to increased rates of glucose import because PEP is used by the phosphotransferase system of *E. coli* to import glucose (see https://doi.org/10.1002/bies.20629). The current study has confirmed the higher PEP levels, which is consistent with this model.

We thank the reviewer for pointing out this missed opportunity. We have expanded the discussion around the role of pykF mutations and the elevated concentrations of PEP observed in our data in section 3.4.

In the introduction, the papers cited to show the importance of changes in metabolism for adaptation do not seem to fit the focus of this study very well. They stress production of toxins and secondary metabolites, which do not seem to be mechanisms that are at work in the LTEE. I can think of two areas of background that would be more relevant: (1) studies of how bacterial metabolism evolves in adaptive laboratory evolution (ALE) experiments to optimize metabolic fluxes toward biomass production (for example, https://doi.org/10.1038/nature01149), and (2) discussions of how cross-feeding, metabolic niche specialization, and metabolic interdependence evolve in microbial communities, including in other evolution experiments (for example, https://doi.org/10.1073/pnas.0708504105 and https://doi.org/10.1128/mBio.00036-12).

We thank the reviewer for pointing out missed citations in our introduction. We agree that these papers are relevant to the topic and have added their citations. Additionally, following the suggestion of another reviewer, we have reorganized the introduction so that the concept of the role of metabolism in evolution is presented first and the LTEE second.

**Reviewer #2 (Public Review):**
[...] Overall, this is a significant and well-executed research study. It offers new insights into the complex relationship between genetic changes and observable traits in evolving populations and utilizes metabolomics in the LTEE, a novel approach in combination with RNA-seq and mutation datasets.However, the paper's overall clarity is lacking. It is spread too thin and covers many topics without a clear focus. I strongly recommend a substantial rewrite of the manuscript, emphasizing structure and readability. The science is well executed, but the current writing does not do it justice.

We thank the reviewer for their positive comments and their constructive feedback on the lack of clarity in writing. Following the reviewer’s suggestions, we have rewritten parts of the manuscript and reorganizd a few sections to improve readability. We hope the revised manuscript is significantly improved.

**Recommendations for the authors**

**Reviewer #1 (Recommendations For The Authors):**
1. Title and Abstract: Add the study organism to the abstract, and probably also the title. Currently, *E. coli* is not mentioned in either! I'm also not sure that the LTEE is a sufficiently well-known acronym to abbreviate this in the title.

We have revised the title of the manuscript and now spell out LTEE and included *E. coli* in the title and the abstract.

1. Abstract: I would switch the usage of metabolome to metabolism in a few more places. For example, "changes in its *metabolism*", "networked and convoluted nature of *metabolism*". The metabolome, the concentrations of all metabolites, is what is being measured, but I think of this as a phenotypic readout of how metabolism evolving.

We have changed “metabolome” to “metabolism” in cases where we refer to what is evolving and use “metabolome” when we refer to what is being measured.

1. Line 16: Technically, the 12 LTEE populations were not initially identical. The Ara- differed from the Ara+ ancestors by one intentional mutation and one unintentional mutation that was not discovered until whole genomes were sequenced. I would rephrase this to "where 12 replicate populations of *E. coli* are propagated" or something similar so that it can be correct without needing to describe this unnecessary detail.

The line has been rephrased as suggested.

1. General Note: The text refers to populations as Ara-3 but the figures use A-3. I'd suggest going with A-3 and similar throughout for consistency.

Instances of Ara have been changed to A+/-, and a sentence specifying as such has been added to the intro to make mention of this.

1. Lines 43-44, 97-98. My understanding is that both S and L ecotypes in A-2 can use both glucose and acetate, but that the differentiation is related to their specialization that leads to each one being better on one or the other nutrient. The descriptions make it sound like each grows at a different time. Also, by definition, cells are not growing during "stationary phase". The change from glucose utilization (and acetate secretion) to acetate utilization during one cycle of growth is better described as a diauxic shift.

We have reworded this part to remove mention of “growth” during stationary phase and changed the wording such that it no longer sounds like they grow at different times.

1. Line 54: The statement "provide the ability to test hypotheses from previous data" is vague. Either provide an example or delete.

We have removed this sentence as suggested.

1. Lines 71-72: The terms "interphase" and "intraphase" sound too much like parts of the cell cycle. I'd suggest describing the comparisons as between and within growth phases.

The use of intra and interphase have been changed as suggested.

1. Line 79: The citrate is presumably still a chelating agent, so change phrasing to "Citrate is present in the medium because it was originally added as a chelating agent" or something similar.

This sentence has been rewritten as suggested.

1. Line 83: Write out "mutation accumulations" so it is easier to understand as "the number of mutations that have accumulated".

The phrase has been changed as suggested.

1. Line 116: It's unclear whether the abundances of metabolites are "strategies of survival" in stationary phase. An equally valid explanation is that there is less selection on the metabolome to have a specific composition during stationary phase to have high fitness.

We have added a line about the possibility for alternative hypotheses.

1. Figure 1: There seems to be some information missing from the legend. What are R06 and R07 in Panels A and B? Is panel D exponential phase and panel E stationary phase?

This information was inadvertently missing from the caption and has been added.

1. Figures 2 and 3: Gene names should be in italics. To me, the gray for deleted genes is hard to tell apart from the blue/red. Perhaps you could put a little X in these boxes instead? I think that having a little triangle pointing from each gene or metabolite name its corresponding abundance panel would help the reader track which information goes with which features. In Fig. 3 the placement of L-aspartate is a bit awkward. I'd suggest moving it down so the dashed line does not have to go through the abundance panel.

These figures have been edited to include small triangles that link a gene or metabolite and its heatmap. Additionally, an X has been added where genes have suffered inactivating mutations and the placement of some elements has been moved to improve overall clarity.

1. Lines 183-185: It would be easier to see and judge the consistency of these argR related relationships if a correlation graph of some kind was shown, probably as a supplemental figure. This plot could, for example, have genes/metabolites across the x-axis and fold-change on the y-axis with lines connecting points corresponding to each of the twelve populations across these categories (like Fig S8 but with lines added). Alternatively, it could be a heat map with the populations across one axis and the genes/metabolites across the other axis (like Fig S3).

We have added a supplementary figure consisting of heatmaps showing the consistency of these changes within an evolved line. It is now figure S9.

1. Line 195: I think adding a sentence elaborating on what exactly mutation accumulation means in this context would be helpful to readers.

We have attempted to clarify the meaning of this by specifically stating that it is due to the accumulation of deleterious mutations.

1. Line 293: Is standard LTEE medium DM25? These omics experiments with the LTEE sometimes use similar media with different glucose concentrations, and this is a very important detail to precisely specify.

We reference “standard” LTEE medium in the methods section and have additionally specified the amount of sugar to make it clear that we are not supplementing the media with additional sugar.

1. Figure S8B. Is "cystine" used instead of "cysteine" on purpose here since the compound is oxidized in the metabolomics treatment?

The use of cystine is intentional, we detect the oxidized compound.

**Reviewer #2 (Recommendations For The Authors):**
Title:The abbreviation "LTEE" should not be in the title. Most readers will not recognize what it means. Instead, either the full name of the experiment, "Long-Term Evolution Experiment with *E. coli*," should be used, or the title should be rephrased to "Linking genotypic and phenotypic changes during a long-term evolution experiment using metabolomics."

We have spelled out LTEE and included *E. coli* in the title.

Abstract:Sentence 1: Consider softening the statement: "Do changes in an organism's environment, genome, or gene expression patterns often lead to changes in its metabolome?"

We have rephrased this sentence to “Changes in an organism's environment, genome, or gene expression patterns can lead to changes in its metabolism”.

Sentence 4: Use a hyphen for "Long-Term."

This addition has been made.

Sentence 4: Replace "transduce" with a more appropriate term: "...how the effects of mutations can be distributed through a cellular network to eventually affect metabolism and fitness."

We have rewritten this sentence as “to understand how mutations can eventually affect metabolism and perhaps fitness”.

Sentence 5: Clarify the use of "both" to refer to the ancestor of the LTEE and its descendant populations as two classes.

We have reworded this sentence so it’s clear that the ancestors and evolved lines are two separate classes “We used mass-spectrometry to broadly survey the metabolomes of the ancestral strains and all 12 evolved lines…”.

Sentence 6: Reverse the order for better emphasis: "Our work provides a better understanding of how mutations might affect fitness through the metabolome in the LTEE, and thus provides a major step in developing a complete genotype-phenotype map for this experimental system."

We have rearranged this sentence per the reviewers suggestion.

Introduction:Revise the introduction for clarity, readability, and logical narrative progression. Start with the second paragraph to set up the basic scientific principles being studied and then transition to describing the LTEE as a model system to examine those principles.

The introduction has been rearranged and reworded in parts to increase clarity.

Sentence 1: Revise for clarity: "The Long-Term Evolution Experiment (LTEE) has studied 12 initially identical populations of *Escherichia coli* as they have evolved in a carbon-limited, minimal glucose medium under a daily serial transfer regime."Sentence 2: Suggestion: "Begun in 1988, the LTEE populations have evolved for more than 75,000 generations, making it the longest-running experiment of its kind."Paragraph 2, sentence 2: Italicize "*Drosophila*."Paragraph 3, sentence 2: Make an important distinction: "Ara-3 is unique in that it evolved the ability to grow aerobically on citrate."Paragraph 3, sentence 4: Introduce the IS-mediated loss of the rbs operon in the LTEE as if it has not been described elsewhere.

These suggestions have been incorporated into the manuscript.

Results:Section 3.1: The use of samples from hours 2 and 24 to represent exponential and stationary phase may present some issues. For instance, capturing Ara-3 during its exponential growth on glucose, but not citrate, at hour 2. Furthermore, except for Ara-3, the LTEE populations reach stationary phase after approximately 4 hours, and there could be significant differences between early, mid, and late stationary phase. This possibility should be acknowledged, and future follow-up work should consider exploring these differences.

We have added sentences in the first paragraph of the results section to include these details. We have also added a short paragraph to the conclusions suggesting additional studies of stationary phase, citing work on evolution of *E. coli* during long term stationary phase.

Paragraph 3: While Turner et al. 2017 is an essential reference regarding resource use differences between Ara-3 and other LTEE populations, it would be more suitable to reference Blount et al. 2012 for the mutations that enabled access to citrate. Also, it is important to note that the difference lies in the ability to grow aerobically on citrate, rather than the ability to metabolize it.

This citation has been added.

Paragraph 4: As mentioned elsewhere, most LTEE populations exhibit balanced polymorphisms. Therefore, it is more appropriate to state that Ara-2 is the best-understood example of long-term diversity. It is likely that there are important metabolic differences between co-existing lineages in other LTEE populations.

We now refer to Ara-2 as being the best-understood example of long term diversity..

Paragraph 5: The first sentence of this paragraph should likely end with "levels."

The word “levels” was added to the end of this sentence.

Figure 3: It is preferable to refer to the "Superpathway of arginine and polyamine biosynthesis," citing EcoCyc as a reference, rather than a descriptor.

This has been changed to a reference.

Section 3.3, Paragraph 3: While higher intracellular amino acid abundances may facilitate higher translation rates and faster growth, the higher abundances themselves do not evaluate the hypothesis. To evaluate the hypothesis, it is necessary to demonstrate that higher abundances are associated with higher translation or growth rates. Therefore, the final sentence of this paragraph is not meaningful.

We have reworded this sentence to say that it’s not possible to tell what the additional amino acids are being used for given only this data and that additional experiments are needed to confirm this hypothesis.

Section 3.4: The first paragraph of this section misstates how evolution works. The low level of glucose in the LTEE does not drive innovation; instead, innovation occurs at random through the introduction of variation by mutation. Although the existence of the citrate resource acts as a reward that selects for variation that provides access to it, it is essential to remember that evolution is blind to such a reward. Moreover, regarding the evolution of the Cit+ trait, it is incorrect to assert that low glucose contributed to its evolution. As shown by Quandt et al. (2015), it seems probable that Cit+ evolution was potentiated by adaptation to specialization on acetate, which is produced by overflow metabolism resulting from rapid growth on glucose. This rapid growth only occurs when glucose is relatively abundant. The level of glucose seems low to us because it is low relative to traditional levels in bacteriological media, but not to the bacteria.

We agree that this is a semantical, but important distinction. We have reworded this part as to not suggest that evolution has any forward thinking properties and is indeed blind to any rewards that might occur as the result of adaptation.

In general, all instances of "utilize" and its cognates should be replaced with "use" and its cognates.

Instances of “utilize” have been changed to use and its cognates.

There is some uncertainty about the expectation of ramping up the TCA cycle in the LTEE. Overflow metabolism and acetate production appear to be prevalent in the LTEE, suggesting that many lineages only partially oxidize carbon derived from glucose, thereby bypassing the TCA cycle. While it is possible that this interpretation is incorrect, it would be helpful to see it addressed in the manuscript.

We agree that this is a plausible hypothesis, we have added a paragraph at the end of this section that discusses the implications of overflow metabolism as an alternative hypothesis.